# Molecular Identification and Engineering a Salt-Tolerant GH11 Xylanase for Efficient Xylooligosaccharides Production

**DOI:** 10.3390/biom14091188

**Published:** 2024-09-20

**Authors:** Jiao Ma, Zhongke Sun, Zifu Ni, Yanli Qi, Qianhui Sun, Yuansen Hu, Chengwei Li

**Affiliations:** 1School of Biological Engineering, Henan University of Technology, Zhengzhou 450001, China; 2Food Laboratory of Zhongyuan, Luohe 462333, China; 3College of Life Science, Henan Agriculture University, Zhengzhou 450046, China

**Keywords:** salt-tolerant GH11 xylanases, thermostability, catalytic activity, site-directed mutagenesis, xylooligosaccharides

## Abstract

This study identified a salt-tolerant GH11 xylanase, Xyn^st^, which was isolated from a soil bacterium *Bacillus* sp. SC1 and can resist as high as 4 M NaCl. After rational design and high-throughput screening of site-directed mutant libraries, a double mutant W6F/Q7H with a 244% increase in catalytic activity and a 10 °C increment in optimal temperature was obtained. Both Xyn^st^ and W6F/Q7H xylanases were stimulated by high concentrations of salts. In particular, the activity of W6F/Q7H was more than eight times that of Xyn^st^ in the presence of 2 M NaCl at 65 °C. Kinetic parameters indicated they have the highest affinity for beechwood xylan (*K_m_* = 0.30 mg mL^−1^ for Xyn^st^ and 0.18 mg mL^−1^ for W6F/Q7H), and W6F/Q7H has very high catalytic efficiency (*K_cat_*/*K_m_* = 15483.33 mL mg^−1^ s^−1^). Molecular dynamic simulation suggested that W6F/Q7H has a more compact overall structure, improved rigidity of the active pocket edge, and a flexible upper-end alpha helix. Hydrolysis of different xylans by W6F/Q7H released more xylooligosaccharides and yielded higher proportions of xylobiose and xylotriose than Xyn^st^ did. The conversion efficiencies of Xyn^st^ and W6F/Q7H on all tested xylans exceeded 20%, suggesting potential applications in the agricultural and food industries.

## 1. Introduction

Xylanases (EC 3.2.1.8) are a kind of enzyme that degrades xylan to xylooligosaccharides (XOS) and a small amount of xylose [1]. They are classified into a number of glycoside hydrolase (GH) families, with the vast majority of characterized xylanases belonging to the GH10 and GH11 families [2]. The GH11 xylanases are the most typical and true xylanases due to their substrate specificity toward xylan and only contain one catalytic domain [3]. These enzymes are conserved in structural domains, e.g., resembling a partially closed right hand and comprising two twisted β-sheets as well as a single α-helix. Stable Fingers and Palm structures are formed when the hydrophobic surfaces of two β-sheets accumulate together. Both the activation and binding sites are hidden in cracks in the Palm structure [4]. Amino acid residues at binding sites participate in substrate binding and recognition through hydrogen bonding or hydrophobic stacking interactions [5]. Between the two β-chains of the β-fold layer, there is an 11-amino acid-long ring shaped like a right-handed “thumb”. The binding of the substrate causes a change in the conformation of the thumb, and the exact position of the “thumb” determines the width of the catalytic crack, which plays a key role in substrate selectivity [6]. Currently, GH11 xylanases with different characters are required to satisfy industrial applications. In particular, salt-tolerant xylanases became a demand for many industrial applications, e.g., high-salt and marine food processing, aquatic feed production, industrial wastewater treatment, saline–alkali soil improvement, and global carbon cycling [7].

Catalytic activity and stability are the two most critical characteristics for xylanases [8]. Improving the catalytic properties and thermostability of GH11 xylanases on the basis of salt tolerance appears to have more practical value. The engineering of GH11 xylanases by directed evolution and rational design is gaining increasing attention [9]. Many directed evolution methods, including DNA shuffling, error-prone polymerase chain reactions, saturation mutagenesis, and disparity mutagenesis, were used in the irrational design of xylanases. Though robust and effective, these strategies are time-consuming and labor-intensive due to the heavy workload during screening [10]. On the other hand, rational design based on structure and function is somewhat more efficient for molecular modification of xylanase [11]. The replacement of some key amino acids often greatly changes the function of the protein [12]. Site-directed mutagenesis (SDM) of these key amino acids is an effective method to improve GH11 xylanase performance [13].

As proved through experimental validation, *Bacillus* sp. have greater and vital xylanolytic exertion with thermophilic advantage [14]. According to the CAZy database, there are already 28 xylanases cloned from *Bacillus* sp. that have been characterized (http://www.cazy.org/GH11_characterized.html, accessed on 7 September 2024). Among them, *B. subtilis*, *B. pumilus*, *B. licheniformis*, and *B. amyloliquefaciens* are widely recognized hosts. Notably, the majority structures in the Protein Data Bank (PDB) come from unclassified *Bacillus* sp. xylanases. *B. subtilis* XynA, one of the representative GH11 xylanases, has a relative constant activity at 40 °C and peaks at pH 7.0 [15]. In contrast, the optimal pH of *B. amyloliquefaciens* XynA is 4, and the optimal temperature is 25 °C [16]. In addition, the optimal pH of XynA from *B. subtilis* VSDB5 and *B. licheniformis* KBFB4 is 6, and the optimal temperature is 50 °C [17]. Other xylanases, like XynBYG from *B. pumilus* BYG, Bcx from *B. circulans*, and BaxA from *B. amyloliquefaciens*, were also reported with different properties [11,18,19]. From a review, the activities of most *Bacillus* wide-type xylanases are generally low [20]. Some of them have also been muted to improve catalytic activity and stability via protein engineering based on sequence and structure [12,13]. Although *Bacillus* are major industrial workhorses and enzymes cloned from them share about 50% of the enzyme market, *Bacillus* xylanases face limitations such as structural destabilization and lower activation energy at extreme temperatures [21,22].

In this study, we aimed to improve the catalytic performance and thermostability of a salt-tolerant GH11 xylanase. Firstly, we cloned and expressed the salt-tolerant GH11 xylanase gene from *Bacillus* sp. SC1. Secondly, we screened mutant enzymes with improved catalytic activity and thermostability by SDM under sequential and structural guidance. Thirdly, the differences between wild-type and mutant enzymes in catalytic activity, thermostability, substrate binding performance, kinetic parameters, and salt tolerance were analyzed. Finally, we analyzed possible reasons for these improved characters by molecular dynamic simulation and tested their applicability for producing XOS using different xylan resources.

## 2. Materials and Methods

### 2.1. Strains and Reagents

The salt-tolerant xylanases gene was obtained from *Bacillus* sp. SC1 (a strain isolated from the soil in 2017 and stored in our laboratory). Bacterial cells of *Escherichia coli* DH5α (*E. coli* DH5α) were used for cloning, and *E. coli* BL21 (DE3) cells were used for protein expression. Taq polymerase, a DNA gel extraction kit, isopropyl-β-D-thiogalactopyranoside (IPTG), and high-fidelity DNA polymerase were purchased from Takara (Takara Biomedical Technology Co., Ltd. Beijing, China). The one-step cloning DNA ligase was ordered from Vazyme (Vazyme International LLC, Nanjing, China). Beechwood xylan, bagasse xylan and corncob xylan were purchased from a commercial provider (Yuanye Co. Ltd., Shanghai, China). Wheat straw xylan was prepared from wheat straw by alkaline extraction, as described elsewhere [23]. Standards of xylose (X), xylobiose (X2), xylotriose (X3), xylotetraose (X4), and xylopentaose (X5) with purity ≥ 98% were purchased from Aladdin (Aladdin Biochemical Tech. Ltd., Shanghai, China). Kanamycin (Kan) sulfate powder was ordered from Sigma (Sigma-Aldrich, Shanghai, China).

### 2.2. Xylanase Activity Assay

Xylanase activity was analyzed according to the method reported elsewhere [24]. The reaction mixture containing 0.1 mL of a diluted enzyme solution (either crude extract or purified protein) and 0.9 mL of 10 mg/mL xylan substrate in 50 mM Gly-NaOH buffer (pH 9.0) was incubated at 55 °C for 10 min. The released reducing sugar content was evaluated by the 3,5-dinitrosalicylic acid (DNS) method using xylose as the standard [25]. One unit (U) of enzyme activity was defined as the amount that released 1 μmol reducing sugar equivalents per minute from the substrate under the above-described standard assay conditions. The absorbance at 540 nm was monitored by a Multimode Reader Spark^®^ (Tecan, Groedig, Austria). For each assay, triplicate measurements were conducted to obtain the mean activity value.

### 2.3. Isolation and Identification of a Salt-Resistant Xylanase

The 19 isolates showing obvious degradation of xylan on agar were inoculated into LB broth for fermentation. Overnight culture supernatants were collected as crude enzyme extracts and used for xylanase activity assay. One isolate, *Bacillus* sp. SC1 with the highest xylanase activity, was inoculated into 200 mL LB, agitating at 37 °C for 24 h. A volume of 50 mL saturated ammonium sulphate solution was added into the overnight culture supernatants with stirring, and the mixture was left for 30 min to precipitate large fragments. Then, the supernatant was collected after centrifugation at 1000 RCF for 10 min. A volume of 80 mL saturated ammonium sulphate solution was added into the collected supernatant for further precipitation. After a second-round centrifugation at 1000 RCF for 10 min, the pellet was resuspended in 5 mL 50 mM Gly-NaOH buffer (pH 9.0). The resolved solution containing the target enzyme was used for sodium dodecyl-sulfate polyacrylamide gel electrophoresis (SDS-PAGE) analysis and further identification by liquid chromatography coupled with mass spectrometry (LC-MS/MS).

### 2.4. Cloning of Xyn^st^ and Its Heterologous Overexpression

The mature form of the Xyn^st^ gene was amplified by polymerase chain reaction (PCR) using high-fidelity DNA polymerase and primers P-XF/PX-R (Appendix A), with the genomic DNA extracted from *Bacillus* sp. SC1 as a template. The specific fragment with a size of 558 bp and the expression plasmid pET-28a(+) were digested with *Xho*I and *Hind*III, respectively. Then, the ligate was transformed into *E. coli* DH5α for cyclization with the ClonExpress II One Step Cloning Kit and subcloned into *E. coli* BL21 for expression. To facilitate purification via affinity chromatography, Xyn^st^ was fused to a hexa-histidine (6×His) tag at the C terminus. The positive transformants grown on antibiotic agar (50 μg/mL Kan) were screened by colony PCR. After sequencing validation, a single clone with correct ORF was propagated in LB medium containing 50 μg/mL Kan, until OD_600_ reached 0.6 (37 °C, 180 rpm). Expression was induced by the addition of 0.8 mM IPTG for 16 h at 25 °C [26].

### 2.5. Rational Designing and Xylanase Mutant Library Construction

To improve the thermostability and hydrolysis characteristics of Xyn^st^, SDM libraries were constructed at eight sites (Y5, W6, Q7, W9, Y65, R49, R112, and W129) by computer docking between xylanase and xylohexaose/xylotriose using Autodock software (version: 4.2.6). Eight pairs of primers (Appendix A) were used for PCR, using the plasmid pET-28a-Xyn^st^ as a template. The complete plasmid was amplified and then digested by restriction enzyme *Dpn*I. The mutant plasmids were introduced into *E. coli* BL21 as in a previous report [12]. In order to enhance mutants further, the mutation sites determined in the first round were subjected to a second-round construction for combinatorial mutagenesis. The xylanase genes in all colonies with improved activities were amplified by PCR. The PCR fragments were sent out for nucleotide sequencing (General Biol. Co. Ltd., Chuzhou, China) to confirm the mutations and potential changes in corresponding amino acids.

### 2.6. High-Throughput Screen of Mutant Library

To screen the SDM libraries, a high-throughput method was developed. Briefly, all colonies were picked into deep-well microtiters that were filled with 0.6 mL sterile LB broth containing 50 μg/mL Kan in each well. After agitating at 180 r/min at 37 °C, IPTG solution was added to a final concentration of 0.8 mM when optical density at 600 nm (OD_600_) reached 0.6. Then, the temperature was shifted to 25 °C for protein expression induction (16 h). A volume of 0.2 mL bacterial culture from each well was transferred to a new 96-well microtiter plate for the collection of cell pellets. After rinsing twice, the cell pellets were resuspended in 0.2 mL 50 mM Gly-NaOH buffer and disrupted by a Non-Contact Ultrasonic Crusher (Jingxin XM08-II, Shanghai, China) at 4 °C for 30 min (Capacity 100%, Turn on 10 s, Pause 10 s). After centrifugation at 8000 RCF for 5 min, 50 µL supernatants were transferred into a 96-well PCR microtiter, and supplemented with 100 µL bagasse xylan solution (10 mg/mL). The microtiter was placed into a vibrated pre-heated Dry Bath Incubator (FOUR E’S Mixer, Guangzhou, China) for enzymatic reaction. Ten minutes later, 50 µL DNS solution was added immediately and heated at 99 °C in a Thermal Cycler (Eppendorf ^®^nexus, Hamburg, Germany) for 15 min. On each microplate, *E. coli* BL21/pET-Xyn^st^ was inoculated as a control during library screening. After detection, colonies with significantly improved absorbance were propagated for three consecutive generations.

### 2.7. Recombinant Xylanases Purification, Quantification, and Electrophoresis

The crude enzyme solution was collected through the following steps. Bacterial cells were collected by centrifugation (8000 RCF, 5 min, 4 °C) and suspended in 50 mM Gly-NaOH buffer (pH 9.0). Then, they were ultrasonicated at 4 °C to disrupt cells completely (10 s on and 10 s off, lasting for 30 min). Subsequently, the ultrasonicated supernatant and insoluble fraction were separated by centrifugation at 8000 RCF for 5 min. The supernatant was mixed with high-affinity Ni^2+^-charged resin (L00223, GenScript Corporation, Nanjing, China) and then incubated for 1 h by gently inverting it in an ice bath to allow the protein to bind to the resin. The slurry was transferred into a column, and the column was washed with 8 bed volumes of wash buffer (20 mM Tris, 500 mM NaCl, 40 mM imidazole, adjust pH to 7.4 using HCl) and 10 bed volumes of elution buffer (20 mM Tris, 500 mM NaCl, 250 mM imidazole, adjust pH to 7.4 using HCl). The eluate was collected step by step for subsequent analysis [27]. The enzyme concentration was measured using the Bradford Protein Assay Kit with bovine serum albumin as the standard. The purity of the enzymes was determined by electrophoresis using 15% SDS-PAGE gels [28].

### 2.8. Biochemical Characterization of Xyn^st^ and Its Mutant W6F/Q7H

The optimal pH of xylanases was measured at 55 °C using 50 mM Na_2_HPO_4_-C_6_H_8_O_7_ buffer (pH 3.0–8.0), 50 mM Gly-NaOH buffer (pH 9.0–10.0), and 50 mM Na_2_HPO_4_-NaOH buffer (pH 11.0–12.0). The pH stability of xylanases was determined by incubating it in buffers from pH 3.0 to 12.0 at 25 °C for 8 h and then measuring the residual enzyme activity under standard assay conditions [29].

The optimal temperature of xylanases was measured under temperatures ranging from 30 °C to 90 °C at the optimal pH. The thermostability of xylanase was determined by incubating the enzyme at temperatures ranging from 55 °C to 65 °C for 0–360 min and then measuring the residual enzyme activity [30].

The effect of ions (NaCl, NH_4_Cl, KCl, ZnCl_2_, NiCl_2_, CuSO_4_, MnCl_2_, CaCl_2_, CoCl_2_, LiCl, and MgCl_2_) and additive SDS on enzyme activity was tested as described elsewhere [31]. All chemicals were separately added to a final concentration of 10 mM, and the mixtures were incubated at 25 °C for 1 h. The residual activities were detected under the standard conditions. All experiments were performed in triplicate and non-preincubated enzymes were used as controls.

### 2.9. Test of Substrate Specificity and Kinetic Parameters

The substrate specificity of xylanases was measured using different substrates (beechwood xylan, bagasse xylan, corncob xylan, wheat straw xylan, arabinoxylan, cellulose, starch, 1% *w*/*v*) in 50 mM Gly-NaOH buffer (pH 9.0) at 55 °C [32]. Released reducing sugar content was determined by the DNS method. The kinetic parameters of xylanases were measured with different concentrations (0–10 mg/mL) of xylans under optimal reaction conditions [33]. The constants of *V_max_*, *K_m_*, and *K_cat_* of these enzymes were calculated according to the Michaelis–Menten equation using Origin 2018.

### 2.10. Effect of Salts on Xyn^st^ and Its Mutant W6F/Q7H

The effect of salts on xylanase activity was studied under standard assay conditions with different concentrations of NaCl (0–4 M) or KCl (0–3 M). Relative activities were calculated by setting the activity in the absence of salts as 100%. The effect of salts on xylanase stability were also evaluated. After keeping enzymes in 50 mM Gly-NaOH buffer (pH 9.0) containing 2 M NaCl for 1–6 h, the residual enzyme activity was determined under standard assay conditions. Residual activities were calculated by setting the initial activity of Xyn^st^ as 100%. All experiments were performed in triplicate.

### 2.11. Molecular Dynamics Simulations of Xyn^st^ and Its Mutant W6F/Q7H

Using xylanase (PDB ID: 2dcy) as the template, homology modeling was performed to obtain a three-dimensional structure of Xyn^st^. The structure images of mutant W6F/Q7H were prepared using Pymol (version: 2.5, Schrodinger, LLC, New York, NY, USA). Molecular dynamics (MD) simulations were performed to further assess the global stability of wild-type Xyn^st^ and mutant W6F/Q7H using GROMACS (version: 2019.6 release) at 333 K (60 °C) for 50 ns. Moreover, MD simulations of W6F/Q7H in 0 M and 2 M NaCl ware performed at 65 °C for 50 ns. The AMBER99SB force field was used, and the treated protein was placed inside an SPC water model in a cubic box. Solvent molecule modeling was performed using the TIP3P water model. Na^+^ and Cl^−^ were used for the neutralization of the simulation system. Minimizations were executed with the steepest descent integrator until the maximum force was below 1000 kJ mol^−1^ nm^−1^ on each atom. The bonds linked with hydrogen atoms were restrained with a linear constraint solver. The pressure during simulation was fixed at 1.0 atm using a Parrinello–Rahman barostat with a coupling constant of 2 ps. Simulations were performed with a 2 fs time step. All MD simulations were firstly equilibrated under a constant-volume ensemble for 100 ps and then under a constant-pressure ensemble for 100 ps, and the MD simulation procedures were carried on for 50 ns. The Root Mean Square Deviation (RMSD) and Root Mean Square Fluctuation (RMSF) were calculated using the standard tools of the GROMACS package [34,35].

### 2.12. Hydrolysis Characteristics of Xyn^st^ and W6F/Q7H

Xylose (X) and standards of XOS with different polymerization degrees (X2–X5) were dissolved in pure water and used for analysis by thin-layer chromatography (TLC) and high-performance liquid chromatography (HPLC) separately. The hydrolysis characteristics of Xyn^st^ and W6F/Q7H were evaluated, as described previously [34]. In brief, 0.01 mg/mL xylanase was mixed with 10 mg/mL of substrates (wheat straw xylan or bagasse xylan) and incubated in 50 mM Gly-NaOH buffer (pH 9.0) at 55 °C for specific intervals. The hydrolysis products were analyzed qualitatively by TLC and quantitatively by HPLC [36]. For HPLC analysis, 10 µL hydrolysate was injected into a ZORBAX Carbohydrate Analysis column (4.6 mm ID × 150 mm, 5 µm) with a differential refractive index detector (RID). The temperatures of the chromatographic column and RID were 30 and 40 °C, using 75% acetonitrile as the mobile phase (1 mL/min). All experiments were performed in triplicate.

### 2.13. Data Analysis and Statistics

All original data were expressed as mean ± standard deviation (SD) obtained from at least triplicated experiments. Statistical analysis was performed by using SPSS 19.0 (IBM Corporation, Somers, NY, USA). Significant differences were analyzed by *t*-test and Tukey’s one-way analysis of variance (ANOVA) when necessary. Values of *p* < 0.05 were considered as statistically significant.

## 3. Results and Discussion

### 3.1. Identification and Expression of the Salt-Tolerant GH11 Family Xylanase Xyn^st^

In order to effectively convert xylan into more value-added products, more xylanases with different properties are needed for versatile applications [37]. In a preliminary screen, the highest xylanase activity (45.94 U/mL) was demonstrated in the culture supernatant of a soil isolate, *Bacillus* sp. SC1 (Appendix A). As only xylanases belonging to the GH11 family are considered true xylanases, we need to confirm the classification of this enzyme. The purification of culture supernatants by ammonium sulfate precipitation (65%) obtained active fractions. Loading these fractions on SDS-PAGE showed a band with an MW of ~25 kDa (Figure 1A). Further LC-MS/MS analysis suggested the protein has the highest identity to *B. halotolerans* glycoside hydrolase family 11 protein (sequence coverage 41%, Figure 1B and Appendix A). In fact, xylanase activity in *B. halotolerans* has been reported recently [38]. However, there was no sequence released in the report, and the activity is low (e.g., 23.47 U/mL after fermentation optimization).

Analysis of the reference protein sequence (Accession: WP_101863413) indicates it has a Sec/SPI type signal peptide and a cleavage site at AA28/29 (Figure 1C). According to computational prediction, the protein is relatively stable and hydrophilic (Appendix A). Based on the reference gene, we cloned its mature form into pET-28a and sequenced the corresponding nucleotide sequence (Figure 1D). The BLASTn analysis indicates the cloned gene has more than 95% identity to *B. halotolerans* xylanase (Appendix A). However, the gene is different from XynA, which is one of the most widely studied GH11 xylanases that has been frequently cloned from different genera of *Bacillus* [17]. Heterologous overexpression in recombinant *E. coli* BL21 obtained pure protein (Figure 1E), and 1,4-beta-xylanase activity can be detected in the crude extracts under both normal and high-salt conditions (Figure 1F). Interestingly, the activity is higher under 2 M NaCl, which suggests the enzyme we named Xyn^st^ is a salt-tolerant xylanase. Xyn^st^ is more tolerant to salt than Xyn40, a previously reported salt-tolerant xylanase from marine isolate that has the highest activity at 0.5 M NaCl [39]. In soils and marine sediments, salt can accumulate to high concentrations. To keep replication, microorganisms may need to have enzymes that can resist such extreme conditions. Taken together, these data indicate that Xyn^st^ isolated from strain SC1 is a salt-tolerant GH11 family xylanase.

### 3.2. Molecular Modification, Mutant Screen, and Characterization

Though the activity of Xyn^st^ is higher than many genes cloned from bacteria, it is much lower than a xylanase cloned from *Aspergillus niger* VTCC 017 [40]. To improve the enzyme performance, we constructed SDM libraries based on rational design. As the active and substrate binding sites are crucial for enzymes, we selected eight amino acids (Y5, W6, Q7, W9, Y65, R49, R112, and W129) as targets for saturation mutation based on protein structure and molecular docking results by Autodock. Facilitated by a newly developed high-throughput screening method, we obtained five positive mutants that possessed improved activity after the screening of 1520 colonies in the SDM library and reaction at 65, 75, and 80 °C, respectively (Figure 2A,B and Appendix A). The increase in the specific activity of these mutants was further confirmed using purified proteins (Appendix A). Among them, two adjacent mutations of W6F and Q7H significantly improve enzyme activity. Combination mutation of them increases activity to 6926 U/mg, which is 3.28-fold that of the wide-type enzyme (Figure 2C). The sequencing of the plasmid validates the mutation of two amino acids as designed (Figure 2D).

Further characterization of these two enzymes was conducted with heterologous overexpressed pure proteins (Appendix A). As demonstrated, the optimal temperature of W6F/Q7H is 65 °C, which is 10 °C higher than that of the wide-type Xyn^st^ (Figure 3A). This value of optimal temperature was the same as that exhibited by xylanases from *B. subtilis* and *Orpinomyces sp.* PC-2, but lower than *Thermotoga thermarum* Xyn10B which has an optimum temperature of 80 °C [41,42,43]. W6F/Q7H also obtained improved thermostability as it is relatively stable after incubation at 60 °C (93% residual activity), compared to only less than 65% residual activity for Xyn^st^ after incubation at 55 °C for 360 min (Figure 3B,C). In addition, Xyn^st^ and W6F/Q7H displayed the same optimal pH, with the maximum activity at pH 9.0 (Figure 3D). These two enzymes are relatively stable within 4 h under pH 9, while W6F/Q7H possessed higher residual activity after incubation for 8 h (Figure 3E).

Positively charged ions, including 10 mM K^+^, NH_4_^+^, Mg^2+^, and Ca^2+^, had a stimulating effect on the enzyme activity of Xyn^st^ and W6F/Q7H (Figure 3F). These ions might bind to enzymes directly or indirectly by changing the water activity of the solvent [44]. Conversely, other ions or compounds have inhibitory effects on enzyme activity, which may be due to the complexation of these metal ions with the reaction groups of enzymes [45]. Similar to the previous report of xylanase, Cu^2+^ completely inactivated Xyn^st^ and W6F/Q7H [46]. The inhibition may be due to the interaction of Cu^2+^ with the SH or carboxyl group of the protein, which leads to conformation changes and, subsequently, results in the inactivation of the enzyme [47]. Noticeably, it seems W6F/Q7H is more resistant to Zn^2+^, Mn^2+^, and Co^2+^ but more sensitive to SDS.

Regarding substrate preference, both Xyn^st^ and W6F/Q7H effectively degraded beechwood xylan, bagasse xylan, corncob xylan, and wheat straw xylan, but not arabinoxylan, cellulose, or starch (Appendix A). Further enzyme kinetic analysis with different amounts of substrates confirmed that both enzymes have relatively high affinity, as all *K_m_* constants are less than 1 mg/mL, with the highest being beechwood xylan. For all substrates, W6F/Q7H has higher *V_max_* and *K_cat_* values but lower constants of *K_m_* (Table 1). In particular, site mutations led to a significant increase in *K_cat_*/*K_m_* values (2.03~4.78-fold) compared to that of Xyn^st^. The data demonstrated that our enzymes have higher catalytic efficiencies than other thermophilic xylanases [48]. However, they are lower than a xylanase and its derivatives cloned from *Orpinomyces* sp. PC-2 that had extremely higher *K_cat_*/*K_m_* values [49]. Anyhow, these dynamic parameters clearly showed the superiority of W6F/Q7H over Xyn^st^.

### 3.3. Evaluation of Enzyme Activity and Stability in the Presence of Salt

As Xyn^st^ is a salt-tolerant xylanase, we further evaluated the effects of different salts on catalytic activity and thermostability. NaCl and KCl have no impact on enzyme activity at low concentrations (0–0.1 M), but they remarkably stimulate activity when salt concentrations exceed 0.25 M. As demonstrated, Xyn^st^ and W6F/Q7H exhibit maximal activity in 2 M NaCl (relative activity of 396% and 810%, respectively) (Figure 4A,B). The specific activity reached 16,854.4 U/mg for W6F/Q7H, which is far higher than a salt-tolerant GH11 xylanase derived from *Phoma* sp. MF13, e.g., a maximal activity of 1322.8 U/mg [49]. When the salt concentration exceeded 2 M, the specific activity decreased, agreeing with XynA which is derived from *Zunongwangia profunda* [50]. Increased enzyme activity in a high-salt environment may be due to a low proportion of hydrophobic amino acid residues that are concentrated only in the “palm” to balance the hydrophobic interaction [10]. In addition, 2 M NaCl enhanced the thermostability of Xyn^st^. As demonstrated, the residual activity remained constant in the presence of NaCl, contrasting with a steady decline in the absence of NaCl after incubation at 55 °C for 30 min (Figure 4C). This is helpful for applications when substrates are mixed with salt and need long-term degradation.

### 3.4. Molecular Dynamic Simulations in the Presence and Absence of Salt

According to a molecular modeling analysis of five *Bacillus* GH11 xylanases, *Bacillus* sp. NCL 87-6-10 (sp_NCL 87-6-10) exhibited high thermal stability and achieved a transition state with minimal energy requirements [51]. To explore the possible molecular basis that led to the improved catalytic efficiency and thermostability, MD simulations were performed. Over 90% of residues in the favorable region of the Ramachandran plot indicated that the predicted models are of good quality and suitable for molecular docking and MD simulations (Appendix A). Notably, the RMSD curves of Xyn^st^ exhibit fluctuations within 30 ns but become stable after 40 ns. In contrast, W6F/Q7H displays lower RMSD values and is relatively stable (Figure 5A). For RMSF values, residues 11–18 and 85–90 of W6F/Q7H are significantly lower, suggesting improved rigidity in this region. Oppositely, residues 155–160 of W6F/Q7H are significantly higher, suggesting improved flexibility in this region (Figure 5B). According to Rg analysis, the value after mutation was slightly lower than that of the wild enzyme, which also indicated that the overall structure of the mutant enzyme protein was more compact (Appendix A). The average RMSDs of W6F/Q7H reduced by 0.013 nm (11.6%) under 2 M NaCl, indicating conformational changes (Figure 5C). However, the average RMSF is reduced by 10.4%, indicating enhanced rigidity of the overall structure of W6F/Q7H in the presence of 2 M NaCl (Figure 5D). The 155–160 sites are located at the hinge at the upper end of the alpha helix (Figure 5E). As reported in *Bacillus circulans* GH11 xylanase, increasing the flexibility of the upper hinge of the alpha helix may result in higher collision probability and, therefore, higher catalytic efficiency [11]. The two sites (residues 11–18 and 85–90) with increased rigidity are at the very edge of the enzyme binding pocket, which may account for the improved thermostability of the mutant enzyme (Figure 5E). For example, enhanced thermostability of GH11 xylanase from *Streptomyces rameus* L2001 has been reported due to enhanced N-terminal rigidity and a more compact overall structure [34]. In addition, the grand average hydropathy value (GRAVY) of W6F/Q7H is a bit higher than Xyn^st^ (−0.436 vs. −0.454), suggesting a slight decrease in hydrophilicity [52].

### 3.5. Degradation of Xylans and Production of XOS

To evaluate the applicability of these two enzymes, we performed degradation assays using two kinds of xylan, namely commercial bagasse xylan and self-prepared wheat straw xylan. As demonstrated by TLC, W6F/Q7H always yielded more XOS than Xyn^st^ did (Figure 6A,B). This can also be confirmed by hydrolysate quantification using HPLC (Figure 6C–F). For example, W6F/Q7H released 2.945 mg/mL XOS when using bagasse xylan, and 2.293 mg/mL XOS when using wheat straw xylan as substrate, which is 28.8% and 19.9% higher than that released by Xyn^st^, respectively (Appendix A). Notably, the ratios of X2 released by W6F/Q7H increased along with degradation time, while they turned steady after 2 h for Xyn^st^. Further hydrolysis of standards with different polymerization degrees suggests both Xyn^st^ and W6F/Q7H cannot degrade X2, as no X was detected (Appendix A). In line with the previous report, both Xyn^st^ and W6F/Q7H can degrade X4 and X5 to smaller oligomers but not xylose [53]. Anyhow, in all cases, X2 and X3 are the main hydrolysis products (Appendix A). In particular, X3 has the largest proportion (>50%) in all resulting XOS, which has been reported as one of the most effective bifidogenic components [54]. Within 0.5 h, the conversion rate of W6F/Q7H to XOS can reach 23.1% for wheat straw xylan and 29.5% for bagasse xylan. The data mean W6F/Q7H is superior in the enzymatic production of XOS, comparing the maximum yield of 20.71% from corncob xylan using an acidic xylanase assisted by ultrasound [55]. A small amount of xylose was produced by the hydrolysis of xylan from wheat straw, while no xylose was produced by the hydrolysis of bagasse xylan, which may be due to the different structural characteristics of xylans from different sources [56].

## 4. Conclusions

As the global focus shifts toward using renewable resources like xylan, more effective xylanase are required. The study identified a salt-tolerant GH11 xylanase, Xyn^st^, from a soil bacterium *Bacillus* sp. SC1. A double mutant W6F/Q7H with increased catalytic activity and optimal temperature was obtained by rational designing and high-throughput screening. Both Xyn^st^ and W6F/Q7H xylanases were stimulated by high concentrations of salts. The salt-resistant property of xylanase from *Bacillus* sp. is a novel feature. These enzymes are efficient at degrading different xylans and have substrate specificity. The mutant enzyme W6F/Q7H released more xylooligosaccharides and yielded higher proportions of xylobiose and xylotriose than Xyn^st^ did. In short, the study identified and engineered a novel GH11 xylanase, which shows promising potential in the agricultural and food industries, especially under high-salt conditions.

## Figures and Tables

**Figure 1 biomolecules-14-01188-f001:**
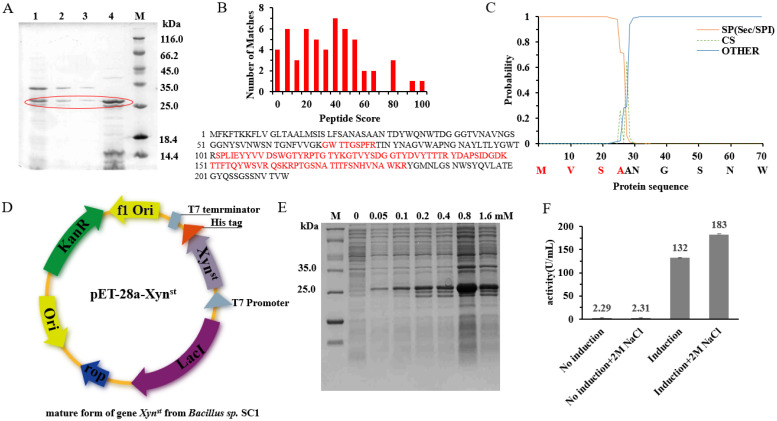
The cloning and identification of a GH11 xylanase from *Bacillus* sp. SC1: (**A**) SDS-PAGE of *Bacillus* sp. SC1 cell free supernatant; (**B**) LC-MS/MS identification of the secreted xylanase; (**C**) prediction of the signal peptide and the cleavage site by SignalP; (**D**) cloning of the mature form of xylanase gene; (**E**) SDS-PAGE of the xylanase expression in recombinant *E. coli* BL21/pET-28a-Xyn at different concentrations of IPTG; (**F**) xylanase activity assay of the cell crude extract of *E. coli* BL21/pET-28a-Xyn. In (**A**), the red cycle indicates the protein bands of the potential target enzyme.

**Figure 2 biomolecules-14-01188-f002:**
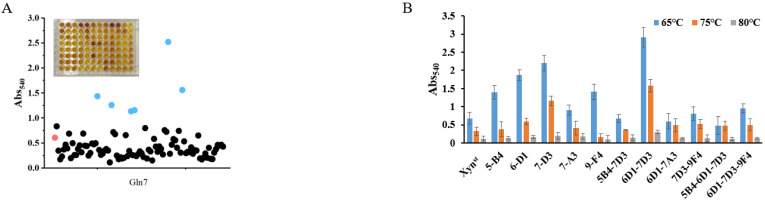
High-throughput screening of SDM libraries for highly active mutants: (**A**) a representative screen of the SDM libraries using 96-well microtiter plate at 65 °C and dot scatter of the absorbance; (**B**) bar chart display of some mutants using cell crude extracts; (**C**) specific activity and relative activity of some mutants using cell crude extracts; (**D**) sequencing of the mutant W6F/Q7H to confirm changes in amino acids. In (**A**), the red dot represents the wide-type strain that is used as control on each microtiter plate. The blue dots are positive mutants that have significantly higher activities, while black dots are negative or nonsense mutants that have activities that are lower than or comparable with the control. In (**C**), the fold line indicates the relative activity. In (**D**), the red box indicates muted nucleotides.

**Figure 3 biomolecules-14-01188-f003:**
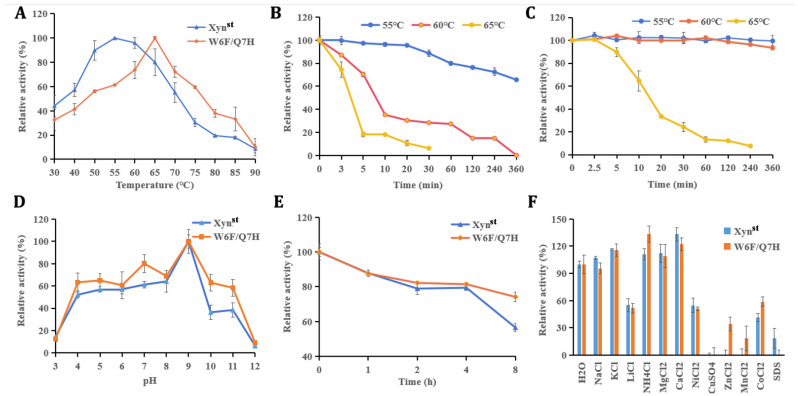
Biochemical characters of Xyn^st^ and W6F/Q7H: (**A**) optimal temperature; (**B**) thermostability of Xyn^st^; (**C**) thermostability of W6F/Q7H; (**D**) optical pH; (**E**) pH stability; (**F**) effects of different chemicals on enzyme activity. The optimum temperature was measured in 50 mM Gly-NaOH buffer at different temperatures (30–90 °C), and the highest enzyme activity was normalized as 100%. The thermostability was tested by incubation of enzymes at 55 °C, 60 °C, and 65 °C for 6 h, respectively. The pH stability was measured at pH 9 for 8 h under 55 °C. For all stability assays, enzyme activities at the beginning before incubation were normalized as 100%, respectively. Data represent the mean ± standard deviation of triplicate measurements.

**Figure 4 biomolecules-14-01188-f004:**
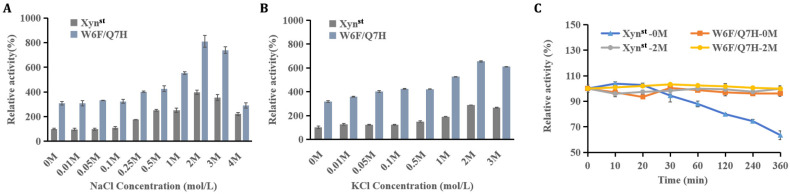
The effects of salts on activity and thermostability of Xyn^st^ and W6F/Q7H: (**A**) effects of NaCl on enzyme activity at different concentrations; (**B**) effects of KCl on enzyme activity at different concentrations; (**C**) effects of NaCl on enzyme thermostability. Relative activities were calculated by setting the activity of Xyn^st^ as 100% in the absence of salts. For enzyme thermostability assays at 55 °C, activities at the beginning before incubation were normalized as 100%, respectively. Data represent the mean ± standard deviation of triplicate measurements.

**Figure 5 biomolecules-14-01188-f005:**
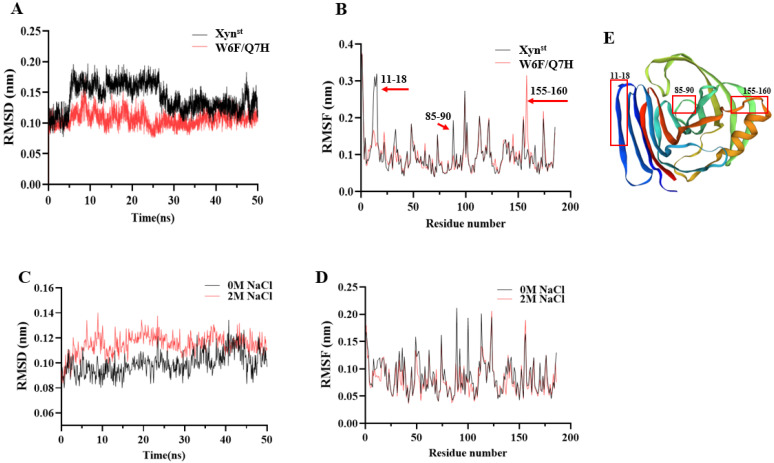
Molecular dynamic simulation of Xyn^st^ and W6F/Q7H: (**A**) the RMSD values of Xyn^st^ and W6F/Q7H within the foremost 50 ns; (**B**) the RMSF values of whole Xynst and W6F/Q7H residues; (**C**) the RMSD values of Xyn^st^ and W6F/Q7H within the foremost 50 ns at 2 M NaCl; (**D**) the RMSF values of Xyn^st^ and W6F/Q7H residues at 2M NaCl; (**E**) predicted molecular structure of W6F/Q7H. RMSD, root mean square deviation, RMSF, root mean square fluctuation.

**Figure 6 biomolecules-14-01188-f006:**
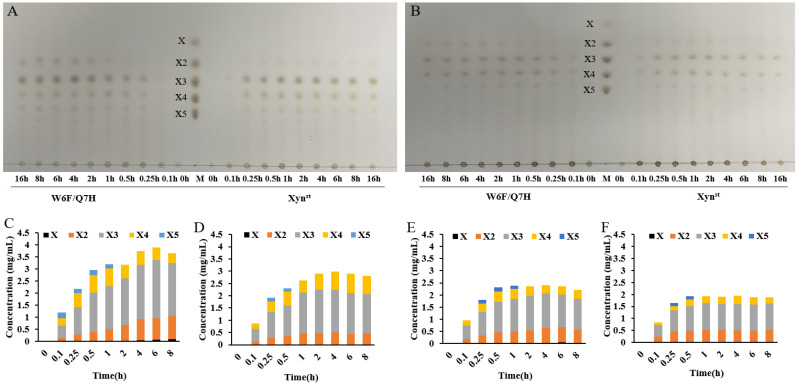
Production of XOS by hydrolysis of xylan using Xyn^st^ and W6F/Q7H: (**A**) TLC analysis of hydrolysate of bagasse xylan after degradation for different hours; (**B**) TLC analysis of hydrolysate of wheat straw xylan after degradation for different hours; (**C**) HPLC quantification of different fractions of XOS after degradation of bagasse xylan by W6F/Q7H for different hours; (**D**) HPLC quantification of different fractions of XOS after degradation of bagasse xylan by Xyn^st^ for different hours; (**E**) HPLC quantification of different fractions of XOS after degradation of wheat straw xylan by W6F/Q7H for different hours; (**F**) HPLC quantification of different fractions of XOS after degradation of wheat straw xylan by Xyn^st^ for different hours. Lane M, xylose and xylooligosaccharides (XOS) standards with different polymerization degrees; X, xylose; X2, xylobiose; X3, xylotriose; X4, xylotetraose; X5, xylopentaose.

**Table 1 biomolecules-14-01188-t001:** Kinetic values and specific activity of Xyn^st^ and W6F/Q7H.

Xylanase	Substrate	*V_max_* (μmol min^−1^ mg^−1^)	*K_m_* (mg mL^−1^)	*K_cat_* (s^−1^)	*K_cat_*/*K_m_* (mL mg^−1^ s^−1^)
Xyn^st^	Wheat straw xylan	2576	0.72	429.33	596.30
	Bagasse xylan	3142	0.84	523.67	623.41
	Beechwood xylan	7123	0.30	1187.17	3957.22
	Corncob xylan	3142	0.84	523.67	623.41
W6F/Q7H	Wheat straw xylan	5488	0.50	914.67	1829.33
	Bagasse xylan	7862	0.44	1310.33	2978.03
	Beechwood xylan	16,722	0.18	2787.00	15,483.33
	Corncob xylan	4941	0.65	823.50	1266.92

## Data Availability

The original contributions presented in the study are included in the article and Appendix A, further inquires can be directed to the corresponding authors.

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
