# Peer review of "Molecular Identification and Engineering a Salt-Tolerant GH11 Xylanase for Efficient Xylooligosaccharides Production"

_biomolecules, 2024, doi:10.3390/biom14091188_

Round 1

Reviewer 1 Report

Comments and Suggestions for Authors

The report submitted by Jiao Ma and co-authors aimed to identify and enhance a salt-tolerant GH11 xylanase, Xynst, capable of withstanding up to 4 M NaCl. Through rational design and high-throughput screening of site-directed mutant libraries, the researchers developed a double mutant, W6F/Q7H, which exhibited a 244% increase in catalytic activity and a 10 °C rise in optimal temperature. Both Xynst and W6F/Q7H were stimulated by high salt concentrations, with W6F/Q7H showing over 8 times higher activity than Xynst in 2 M NaCl at 65 °C. The mutant also demonstrated superior catalytic efficiency and improved structural attributes, leading to higher xylooligosaccharide production and conversion efficiencies, suggesting its potential use in agricultural and food applications.

The study uses a multi-faceted approach, including cloning, expression, site-directed mutagenesis (SDM), and molecular dynamics simulations to enhance and analyze the xylanase. The development of a double mutant (W6F/Q7H) with a 244% increase in catalytic activity and enhanced thermostability demonstrates substantial progress over the wild-type enzyme. The findings suggest practical applications in the agricultural and food industries, particularly in producing xylooligosaccharides (XOS) from various xylan sources, which could have significant commercial implications.

Here are my suggestions for further improving the manuscript.

1.      The study improved thermostability but does not provide data on the long-term stability of the enzyme, which is essential for practical applications in industrial processes.

2.      While the double mutant shows significant improvements, the study might benefit from comparing other potential mutants or modifications to evaluate whether the observed enhancements are optimal or if further improvements are possible.

3.      Change the sentence in lines 101 and 102 to “A volume of 50 mL of saturated ammonium sulfate solution was added to the overnight culture supernatants with stirring, and the mixture was left for 30 minutes to precipitate large fragments.”

4.      Lines 103, 105, and 141, change “centrifuge” to “centrifugation”.

5.      Line 106, replace “resolved” with “resuspended”.

6.      Line 113, the correct notation of Xho I and Hind III is XhoI and HindIII (I and II not italic and no space between Xho and I and Hind and III).

7.      Lines 119 and 136, replace “reaches” with “reached”.

8.      Line 124, write “Autodock” before “software”.

9.      Line 128. Change “was” to “were”.

10. Line 129, change “combinational” to “combinatorial”.

11. Line 138, delete fluids, change “removed into” to “transferred to”, and add “plate” after “microtitor” (in line 142 also).

12. Line 157, change “inverting in an ice bath to bind the protein to the resin” to “inverting it in an ice bath to allow the protein to bind to the resin”.

13. Line 196, “All experiments were analyzed in triplicate.” Analyzing in triplicate and performing/conducting in triplicate are different things. Please clarify whether analysis was done three times for the same experimental set up or the experiments were repeated thrice?

14. Elaborate all the abbreviations used in the manuscript.

15. Write all references in a uniform format, for instance in references 32, 36, and 46 the journal names are written in abbreviated form while in the rest there are full names.

Comments on the Quality of English Language

The manuscript has English language and grammar issues. I have mentioned a few in my comments. Please thoroughly check the manuscript for corrections.

Author Response

Here are my suggestions for further improving the manuscript.

  1. The study improved thermostability but does not provide data on the long-term stability of the enzyme, which is essential for practical applications in industrial processes.

Re:Agree, long-term stability is important for practical applications. In our manuscript, we only reported the stability of this enzyme in solutions during a short period. I would like to reply this comment from two aspects. 1) As showed in Fig. 3B and 3C, the mutant enzyme W6F/Q7H is stable within 6h, contrasting complete lost of activity for the wide-type enzyme at 60°C. Xynst has less than 20% activity after 5min, while W6F/Q7H still has more than 80% activity at 65°C. 2) In practical applications, the enzyme normally used in powder form and hydrolyzed below 60°C within 2h. So, it should be stable enough for industrial applications.

  1. While the double mutant shows significant improvements, the study might benefit from comparing other potential mutants or modifications to evaluate whether the observed enhancements are optimal or if further improvements are possible.

Re:Yes, your suggestion is highly appreciated. We get this double mutant after screening of 1520 colonies. Among these 1520 mutants, we think the double mutant is optimal. Of course, further improvements are possible if other potential mutants or modifications were screened.

  1. Change the sentence in lines 101 and 102 to “A volume of 50 mL of saturated ammonium sulfate solution was added to the overnight culture supernatants with stirring, and the mixture was left for 30 minutes to precipitate large fragments.”

Re:Thank you, the sentence was changed as suggested.

  1. Lines 103, 105, and 141, change “centrifuge” to “centrifugation”.

Re:The word was changed in the revision.

  1. Line 106, replace “resolved” with “resuspended”.

Re:”resolved” was replaced by “resuspended”.

  1. Line 113, the correct notation of Xho I and Hind III is XhoI and HindIII (I and II not italic and no space between Xho and I and Hind and III).

Re:Yes, they were changed as suggested.

  1. Lines 119 and 136, replace “reaches” with “reached”.

Re:The word was changed in the revision.

  1. Line 124, write “Autodock” before “software”.

Re:Corrected as your suggestion.

  1. Line 128. Change “was” to “were”.

Re:Corrected as you suggested.

  1. Line 129, change “combinational” to “combinatorial”.

Re:Corrected as you suggested.

  1. Line 138, delete fluids, change “removed into” to “transferred to”, and add “plate” after “microtitor” (in line 142 also).

Re:Corrected as you suggested.

  1. Line 157, change “inverting in an ice bath to bind the protein to the resin” to “inverting it in an ice bath to allow the protein to bind to the resin”.

Re:Corrected as you suggested.

  1. Line 196, “All experiments were analyzed in triplicate.” Analyzing in triplicate and performing/conducting in triplicate are different things. Please clarify whether analysis was done three times for the same experimental set up or the experiments were repeated thrice?

Re:Thanks for your question. We understand you are wondering whether we performed the experiments for three repeats or just assayed three times using samples obtained from one experiment. For biochemical characterization, the experiments were repeated thrice.

  1. Elaborate all the abbreviations used in the manuscript.

Re:OK. Some abbreviations are commonly used, while others are occasionally presented. To make it clear to undstand, we deleted some of them that are unnecessary.

  1. Write all references in a uniform format, for instance in references 32, 36, and 46 the journal names are written in abbreviated form while in the rest there are full names.

Re:Thank you for your careful reading. We amended these references and checked all other to make sure they are in a uniform format in the revision.

Comments on the Quality of English Language

The manuscript has English language and grammar issues. I have mentioned a few in my comments. Please thoroughly check the manuscript for corrections.

Re:We carefully checked the manuscript. All correction can be found using track-changes on.

Reviewer 2 Report

Comments and Suggestions for Authors

Opinion of the manuscript of a research paper entitled Molecular identification and engineering a novel salt-tolerant GH11 xylanase for efficient xylooligosaccharides production by Jiao Ma.

The manuscript focuses on characterization and engineering of technologically relevant enzyme – xylanase – which is capable of producing xylooligosaccharides from biomasses that are important constituents in food technology and emerging prebiotics. These saccharides have significant potential as functional food ingredient. In the manuscript, the GH11 xylanase from Bacillus sp. SC1 was heterologously expressed and characterized, and structure-function study was conducted using a mutant library. Variant with increased catalytic activity and salt-activation were disclosed that could be used as a catalyst for short xylooligosaccharide production.

The manuscript is comprehensive with many analytical methods used, technologically carefully conducted and mainly clearly written.

Still there are some issues that need to be improved:

1)      There is a high number of GH11-family xylanases described already in the literature. According to CAZy database there are at least 291 enzymes characterized in detail including 42 crystal structures available and many of them from Bacillus sp. The Introduction section is not covering the relevant information regarding the properties of studied and mutated GH11 xylanases what is different and what is similar regarding their biochemical characteristics.

2)      EC number for xylanase is missing from the Introduction.

3)      Title, abstract and elsewhere. It is not clear why the authors have used “novel” excessively (17 times). The type of the activity or the xylanase of Bacillus sp. is not novel at all, but salt-activated feature for xylanases is newly described. The authors should consider wording that is not exaggerating the results.  

4)      The origin of the wild-type enzyme should be mentioned in the abstract.  

5)      It is not clear why the enzyme was named Xynst. The name is not resembling the standard use of xylanase abbreviated names.

6)      The units for maximal velocity and catalytic efficiency should be reviewed. In the current format they are most probably not correct. It would be preferred if brackets are used or -1 style (as in kcat).

7)      Materials and methods. The final concentrations of the components should be used whenever possible.

8)      L110-111. Is the sequence of the gene or genome sequence available in the open databases (NCBI etc)?

9)      L248. The accession refers to the GH11 enzyme of B. halotorerans sequence but the referred Fig. 1. is based on the studied Bacillus sp. SC1. Do the both proteins have signal peptide cleavage at the same location?

10)   L253-254. What is the identity % of the studied enzyme with XynA?

11)   L256. Fig. 1E is not showing purified preparation.  

12)   L272. How identical on the protein level are bacterial and fungal enzymes?

13)   The discussion regarding the enzyme mechanism of action in the context of the organism that harbours the xylanase could be added to the discussion part. What is the reasoning that the organism (Bacillus sp) needs this enzyme tolerating extreme salt content and high temperatures. Overall, the discussion is rather minimal with only selected references used.

Comments on the Quality of English Language

The language could be slightly improved but is understandable and generally correct.

Author Response

The manuscript is comprehensive with many analytical methods used, technologically carefully conducted and mainly clearly written.

Re:Thanks for your positive evaluation of our manuscript.

Still there are some issues that need to be improved:

1)      There is a high number of GH11-family xylanases described already in the literature. According to CAZy database there are at least 291 enzymes characterized in detail including 42 crystal structures available and many of them from Bacillus sp. The Introduction section is not covering the relevant information regarding the properties of studied and mutated GH11 xylanases what is different and what is similar regarding their biochemical characteristics.

Re:You are right. There are already a large number of xylanases described in the literature. Your suggestion is highly appreciated. We added a few sentences to cover the relevant information you are concerned in the Introduction section. Please see line 64-69.

As proved through experimental validation, Bacillus sp. have greater and vital xylanolytic exertion with thermophilic advantage [14]. According to CAZy database, there are already 28 xylanases cloned from Bacillus sp. have been characterized  (http://www.cazy.org/GH11_characterized.html, accessed on 2024.09.07). However, Bacillus xylanases face limitations such as structural destabilisation and lower activation energy at extreme temperatures [15].

By the way, in line 267-270, we have already discussed a bit on its difference from other xylanases cloned from Bacillus sp.. We also added a few sentence to discuss the difference.

Bakry, M.M., Salem, S.S., Atta, H.M., El-Gamal., M. S. & Fouda., A. (2022). Xylanase from thermotolerant Bacillus haynesii strain, synthesis, characterization, optimization using Box-Behnken Design, and biobleaching activity. Biomass Conversion and Biorefinery, 14, 9779–9792. https://doi.org/10.1007/s13399-022-03043-6

Walia, A., Guleria, S., Mehta, P., Chauhan, A., & Parkash, J. (2017). Microbial xylanases and their industrial application in pulp and paper biobleaching: a review. Biotech, 7(1):11. https://doi.org/10.1007/s13205-016-0584-6

Sree Agash, S. G., & Rajasekaran, R. (2024). Exploring Bacillus species xylanases for industrial applications: screening via thermostability and reaction modelling. Journal of Molecular Modeling, 30, 242. https://doi.org/10.1007/s00894-024-06048-2

2)      EC number for xylanase is missing from the Introduction.

Re:This is important. We added the EC number for xylanase (EC 3.2.1.8). Thank you!

3)      Title, abstract and elsewhere. It is not clear why the authors have used “novel” excessively (17 times). The type of the activity or the xylanase of Bacillus sp. is not novel at all, but salt-activated feature for xylanases is newly described. The authors should consider wording that is not exaggerating the results.  

Re:Agree. The type of the activity or the origin of the enzyme is not novel at all. Salt stimulation or induction feature is new. To make the content more objective, we deleted the word “novel” in the manuscript.

4)      The origin of the wild-type enzyme should be mentioned in the abstract.  

Re:OK, we added its origin in the abstract.

5)      It is not clear why the enzyme was named Xynst. The name is not resembling the standard use of xylanase abbreviated names.

Re:Good question. The enzyme has the highest identity to Bacillus halotolerans xylanase, which has been annotated as XynA in NCBI. However, it is still not same to XynA. “st” means salt-tolerant. Therefore, we named the enzyme Xynst by combination of these two features together. By the way, most Bacillus xylanases in the CAZy database were named with Xyn....

6)      The units for maximal velocity and catalytic efficiency should be reviewed. In the current format they are most probably not correct. It would be preferred if brackets are used or -1 style (as in kcat).

Re:Thanks for careful reading. We changed the units using -1 style.

7)      Materials and methods. The final concentrations of the components should be used whenever possible.

Re:Agree. We used the final concentrations when they are possible.

8)      L110-111. Is the sequence of the gene or genome sequence available in the open databases (NCBI etc)?

Re:No. The sequences of this gene and its mutant will be patented.

9)      L248. The accession refers to the GH11 enzyme of B. halotorerans sequence but the referred Fig. 1. is based on the studied Bacillus sp. SC1. Do the both proteins have signal peptide cleavage at the same location?

Re:Yes, the accession refers to the reference sequence of B. halotorerans GH11 enzyme. As described in line 263-264, Fig. 1C is based on the reference sequence, not the studied Bacillus sp. SC1. You asked a good question. In fact, we don’t know whether they have same (predicted) cleavage site. This is because 1) we only get partial peptide fragments after LC-MS/MS identification of the secreted protein from the supernatants; 2) we do not have the genome sequence of strain SC1, and only cloned the mature form based on the corresponding reference gene.

10)   L253-254. What is the identity % of the studied enzyme with XynA?

Re:We suppose you are asking the identity of Xynst and XynA from other Bacillus sp.. We blasted two protein senquences. The identifty between Xynst and XynA (ADK92885.1) from B. amyloliquefaciens is 93.5%.

11)   L256. Fig. 1E is not showing purified preparation.  

Re:Yes, Fig. 1E is the crude extract, just showing the influence of inducer IPTG on the expression of target protein. Purification is unnecessary.

12)   L272. How identical on the protein level are bacterial and fungal enzymes?

Re:This is difficult to answer. In our case, bacterial xylanases and fungal xylanases are quite different on the protein level. For example, the protein identity between Endo-1,4-beta-xylanase A from fungi Vanrija pseudolonga and XynA from bacteria B. amyloliquefaciens is only 14%.

13)   The discussion regarding the enzyme mechanism of action in the context of the organism that harbours the xylanase could be added to the discussion part. What is the reasoning that the organism (Bacillus sp) needs this enzyme tolerating extreme salt content and high temperatures. Overall, the discussion is rather minimal with only selected references used.

Re:OK, we added a few snetences to discuss the reasoning that the organism (Bacillus sp) needs this enzyme tolerating extreme salt content and high temperatures. In soils and marine, salt can accmulate to high concntrations. To keep replication, microorganisms may need to have enzymes that can resist such extreme conditions. Therefore, to effectively degrade xylan, xylanases evolved these features.

Comments on the Quality of English Language

The language could be slightly improved but is understandable and generally correct.

Re:Thanks, we thorough checked the language and made many minor corrections.

Reviewer 3 Report

Comments and Suggestions for Authors

In this paper, the authors identify and describe a new salt-tolerant xylanase that can be used for the degradation of xylans from various sources.

The publication is well structured and clearly organised. Both the identification and the subsequent characterisation are conclusive and well presented.

In order to cope with the large number of possible candidates, a new high throughput method was established, which can also be used for other questions.

Overall, the publication is very well done. Before publication, a few linguistic errors should be corrected and some minor points clarified.

* Publication [14] does not include a xylan preparation, please give another reference.

* line 175: NH4Cl is not a metal ion

* Fig S2 “ … and identified by ??”

* Fig S4 + Fig 2A: Give an explanation for the differently coloured dots

* Fig S8 Please give an explanation why we don´t see X after the degradation of X4. It would be expected X4 -> X2 + X2 or X4 -> X3 + X

* Language:

line 143/144 rephrase sentence

line 162 usING instead of used

line 184 at 55°C

line 185 sugar content

Legend Fig S5 is the target xylanase or are the target xylanases

Comments on the Quality of English Language

minor editing

Author Response

In this paper, the authors identify and describe a new salt-tolerant xylanase that can be used for the degradation of xylans from various sources.

The publication is well structured and clearly organised. Both the identification and the subsequent characterisation are conclusive and well presented.

In order to cope with the large number of possible candidates, a new high throughput method was established, which can also be used for other questions.

Overall, the publication is very well done. Before publication, a few linguistic errors should be corrected and some minor points clarified.

Re:Thanks for reviewing our manuscript.

* Publication [14] does not include a xylan preparation, please give another reference.

Re:Thank you. We replaced this wrong citation with a new reference.

* line 175: NH4Cl is not a metal ion

Re:Yes, this is obviously wrong. We changed it.

* Fig S2 “ … and identified by ??”

Re:We are sorry for this incomplete description. The protein was identified by LC-MS/MS.

* Fig S4 + Fig 2A: Give an explanation for the differently coloured dots

Re:OK, the red dot represents the wide-type strain that is used as control on each microtitor plate. The blue dots are positive mutants that have significantly higher activities, while black dots are negative or nonsense mutants that have lower or comparable activities than the control.

* Fig S8 Please give an explanation why we don´t see X after the degradation of X4. It would be expected X4 -> X2 + X2 or X4 -> X3 + X

Re:This is a good question. We agree that X2 or X3 and X should be obtained after degradation of X4, theoretically. However, we can clearly see that X2 and the majority of X3 can not be degraded futher by xylanase. Therefore, when X4 or X5 was used as substrate, the products are mainly X2 and X3. In fact, no xylose can be detected when bagasse xylan used as substrayte by HPLC, while xylose takes less than 1% when wheat straw xylan was used as substrate (Table S3). Therefore, it is reasonable no visible band of xylose can be seen on TLC gel.

* Language:

line 143/144 rephrase sentence

Re:The sentence was rephrased like “The xylanase genes in all colonies with improved activities were amplified by PCR. The PCR fragments were sent out for nucleotide sequencing (General Biol. Co. Ltd, Chuzhou, China) to confirm the mutations and potential changes of corresponding amino acids.”

line 162 usING instead of used

Re:Changed as suggested.

line 184 at 55°C

Re:Changed as suggested.

line 185 sugar content

Re:Changed as suggested.

Legend Fig S5 is the target xylanase or are the target xylanases

Re:This is a mistake. We changed the sentence like “The bands with a molecular weight of about 26 kDa are target xylanases.”.

Comments on the Quality of English Language

minor editing

Re:Yes, we thoroughly edit the text to avoid any language errors.

Round 2

Reviewer 2 Report

Comments and Suggestions for Authors

The authors have adequately responded to most of the raised issues. There are still two concernes to be addressed:

1) Even though the authors explained where the term Xyn st derived, it is not acceptable. Superscript abbreviations are considered problematic as if the superscript letters are not in the format, it might mean something else. As the authors compared the Bacillus sp. xylalanse with XynA of Bacillus amyloliquefaciens XynA and found it more than 90% identical, it can be concluded that the protein is in fact XynA. If the authors want to stess, it is different variant, XynAB can also be used. (B from Bacillus).

2) The introduction is still too compact and not comprehensive enough thus misses some background information of the properties of Bacillus spp. xylanases.     

Comments on the Quality of English Language

Minor editing still needed.

Author Response

The authors have adequately responded to most of the raised issues. There are still two concernes to be addressed:

  • Even though the authors explained where the term Xyn st derived, it is not acceptable. Superscript abbreviations are considered problematic as if the superscript letters are not in the format, it might mean something else. As the authors compared the Bacillus sp. xylalanse with XynA of Bacillus amyloliquefaciens XynA and found it more than 90% identical, it can be concluded that the protein is in fact XynA. If the authors want to stess, it is different variant, XynAB can also be used. (B from Bacillus).

Re: Thanks for your comment. We used the superscript abbreviation “st” to stress the salt-tolerant feature of the cloned enzyme. However, you think this is problematic and suggest us to name it XynAB. We appreciate your idea. However, XynAB is inappropriate after careful consideration. 1) both xynA and xynB are included in the CAZy database, a mix of them is somewhat misleading; 2) the name XynAB has already been reported in literature, standing for a fusion of Aspergillus niger SCTCC 400264 xylanase genes (Li et al., 2014); 3) We find different names for Bacillus sp. xylanases in the CAZy database, for example, XynA, XynB, Xyn11, XynS, XynY, XylP, XylHB, XylC1, XylC2, XylC3...  (http://www.cazy.org/GH11_characterized.html); 4) Casual naming is widely in acceptable in PubMed, e.g., an ylanase from Bacillus pumilus BYG was named XynBYG (Zhang et al., 2016), Bacillus circulans xylanase was named Bcx (Min et al., 2021), Bacillus amyloliquefaciens xylanase A was named BaxA (Liu et al., 2022). All these information mean there is no consensus standard for naming. Therefore, we think Xynst can be used. Alternatively, we can replace Xynst by XynSC1 if you still think it is unacceptable.

Although the Bacillus sp. SC1 xylanase has more than 90% identity to Bacillus amyloliquefaciens XynA, we do not think they are same. As different characteristics can be seen: the optimal pH of Xynst is 9, and the optimal temperature is 55°C; while the optimal pH of Bacillus amyloliquefaciens XynA is 4, and the optimal temperature is 25°C (Baek et al., 2012). In addition, the optimal pH of xynA from Bacillus subtilis VSDB5 and Bacillus licheniformis KBFB4 is 6, and optimal temperature is 50°C (Joshi et al., 2022). All these indicate they are different enzymes.

Li, X., Zhang, W., Wu, Q., Xu, H., Qiao, D., Cao, Y., & Cao, Y. Construction of fusion protein xynAB and improvement of its pH and thermal stability from Aspergillus niger 400264. J Pure Appl Microbiol. 2014;8(4):2535-2542

2) The introduction is still too compact and not comprehensive enough thus misses some background information of the properties of Bacillus spp. xylanases.     

Re: Agree. We added an extra paragraph to provide enough background information of the properties of Bacillus sp. xylanases.

As proved through experimental validation, Bacillus sp. have greater and vital xylanolytic exertion with thermophilic advantage [14]. According to CAZy database, there are already 28 xylanases cloned from Bacillus sp. have been characterized  (http://www.cazy.org/GH11_characterized.html, accessed on 2024.09.07). Among them, B. subtilis, B. pumilus, B. licheniformis, B. amyloliquefaciencs are widely recognized hosts. Of note, the majority structures in the Protein Data Bank (PDB) come from unclassified Bacillus sp. xylanases. B. subtilis XynA, one of the representative GH11 xylanases, has a relative constant activity at 40°C and peaked at pH 7.0 [15]. In contrast, the optimal pH of B. amyloliquefaciens XynA is 4, and the optimal temperature is 25°C [16]. In addition, the optimal pH of XynA from B. subtilis VSDB5 and B. licheniformis KBFB4 is 6, and the optimal temperature is 50°C [17]. Other xylanases, like XynBYG from B. pumilus BYG, Bcx from B. circulans, and BaxA from B. amyloliquefacien, were also reported with different properties [11, 18, 19]. From a review, the activities of most Bacillus wide-type xylanases are generally low [20]. Some of them have also been muted to improve catalytic activity and stability via protein engineering based on sequence and structure [12, 13]. Although Bacillus are major industrial workhorses and enzymes cloned from them share about 50% of the enzyme market, Bacillus xylanases face limitations such as structural destabilisation and lower activation energy at extreme temperatures [21, 22].

  1. Beliën, T., Joye, I. J., Delcour, J. A., & Courtin, C. M. (2009). Computational design-based molecular engineering of the glycosyl hydrolase family 11 B. subtilis XynA endoxylanase improves its acid stability.Protein engineering, design & selection: PEDS, 22(10), 587–596. https://doi.org/10.1093/protein/gzp024
  2. Baek, C. U., Lee, S. G., Chung, Y. R., Cho, I., & Kim, J. H. (2012). Cloning of a Family 11 Xylanase Gene from Bacillus amyloliquefaciens CH51 Isolated from Cheonggukjang. Indian journal of microbiology, 52(4), 695–700. https://doi.org/10.1007/s12088-012-0260-4
  3. Joshi, J. B., Priyadharshini, R., & Uthandi, S. (2022). Glycosyl hydrolase 11 (xynA) gene with xylanase activity from thermophilic bacteria isolated from thermal springs. Microbial cell factories, 21(1), 62. https://doi.org/10.1186/s12934-022-01788-3
  4. Zhang, W., Yang, M., Yang, Y., Zhan, J., Zhou, Y., & Zhao, X. (2016). Optimal secretion of alkali-tolerant xylanase in Bacillus subtilis by signal peptide screening. Applied microbiology and biotechnology, 100(20), 8745–8756. https://doi.org/10.1007/s00253-016-7615-4
  5. Liu, M., Li, J., Rehman, A. U., Luo, S., Wang, Y., Wei, H., & Zhang, K. (2022). Sensitivity of family GH11 Bacillus amyloliquefaciens xylanase A (BaxA) and the T33I mutant to Oryza sativa xylanase inhibitor protein (OsXIP): An experimental and computational study. Enzyme and microbial technology, 156, 109998. https://doi.org/10.1016/j.enzmictec.2022.109998
  6. Rashid, R., & Sohail, M. (2021). Xylanolytic Bacillus species for xylooligosaccharides production: a critical review. Bioresources and bioprocessing, 8(1), 16. https://doi.org/10.1186/s40643-021-00369-3
  7. Schallmey, M., Singh, A., & Ward, O. P. (2004). Developments in the use of Bacillus species for industrial production. Canadian journal of microbiology, 50(1), 1–17. https://doi.org/10.1139/w03-076